# Functional Acrylic Surfaces Obtained by Scratching

**Abraham Medina** [1,*,†] **, Abel López-Villa** [1,†] **and Carlos A. Vargas** [2,†]

1   SEPI ESIME Azcapotzalco, Instituto Politecnico Nacional, Av. de las Granjas 682,
    Col. Sta. Catarina Azcapotzalco, Mexico City 02250, Mexico; abelvilla77@hotmail.com
2   Departamento de Ciencias Básicas, Universidad Autónoma Metropolitana-Azcapotzalco, Av. San Pablo 180,
    Col. Reynosa Azcapotzalco, Mexico City 02200, Mexico; cvargas@azc.uam.mx
*   Correspondence: amedinao@ipn.mx
†   These authors contributed equally to this work.

**Abstract:** By using sandpaper of different grit, we have scratched up smooth sheets of acrylic to cover their surfaces with disordered but near parallel micro-grooves. This procedure allowed us to transform the acrylic surface into a functional surface; measuring the capillary rise of silicone oil up to an average height $\bar{h}$, we found that $\bar{h}$ evolves as a power law of the form $\bar{h} \sim t^n$, where $t$ is the elapsed time from the start of the flow and $n$ takes the values 0.40 or 0.50, depending on the different inclinations of the sheets. Such behavior can be understood alluding to the theoretical predictions for the capillary rise in very tight, open capillary wedges. We also explore other functionalities of such surfaces, as the loss of mass of water sessile droplets on them and the generic role of worn surfaces, in the short survival time of SARS-CoV-2, the virus that causes COVID-19.

**Keywords:** capillary rise; functional surfaces; human skyn; droplets; COVID-19

## 1. Introduction

Nowadays, research on the fabrication and physical behavior of functional surfaces, e.g., micro-structured surfaces with singular features able provide one or more functional properties, mainly those related to fluid transport, is a very relevant topic of study [1–8].

Commonly, the transport of liquids on functional surfaces is governed by the wetting, capillary action and the gravitational field, and therefore the details of the microstructured and nanostructured surfaces are essential to understand how liquid spreads in the preferred direction [2,5,8–12]. A feature of the wettable functional surfaces that has attracted much attention in recent years for its broad potential applications is that the spreading occurs without energy input, such as non-powered delivery systems, self-lubrication and microfluidic devices [8].

Depending on applications, the texture of many functional surfaces consists of grooves (open capillaries) of different height, width, and contact angle [2,10,11]. For example, in technology, micro V-grooves frequently appear, as in the case of grooved heat pipes where parallel micro grooves were drawn in copper plates to improve their energetic efficiency [2]. In nature, directional spreading of water is a feature that *Nepenthes alata,* a carnivorous pitcher plant whose slippery peristome remains completely wetted by water, has used as the source of an insect capture function [6]. Moreover, the structure of *Nepenthes* has been useful in the fabrication of bio-inspired surfaces through the replica molding method [3–8].

Incidentally, human skin bears similarities to a microfluidic system since the external layer forms a microchannel network comprised of large numbers of interconnected micro V-grooves. For instance, experiments on deposition of oil drops (moisturizer) on the middle of the forearm show that radial flows away from an initially placed drop occur through the grooves [12].

Similarly, functional acrylic surfaces are not new, for instance, acrylic sheets were micro-textured to generate fishbone architecture, to get blood repellent surfaces, by using

continuous wave UV laser, which can generate patterned structures in range of 1–300 microns [13].

In the present work, we are interested in the fabrication and characterization of functional surfaces obtained through the raw scratching up of acrylic sheets, since the resulting irregular surface pattern dramatically modifies their surface wetting properties. Moreover, the issue of soft surfaces scratching is ubiquitous; however, we scarcely pay attention to it in our daily life, despite the fact that many surfaces at home, offices, hospitals, etc., turn into functional surfaces due to its use and necessary cleaning. Both facts produce finely and irregularly scratched surfaces.

To perform the study of the functionality of the scratched surfaces, we will texturize surfaces by following a simple protocol of longitudinal scratching up of acrylic sheets, which allows liquids to sprint uphill through micro V-grooves. In the experiments, we will harness such physical phenomenon, to understand the scope of a fluid mechanics model based on the capillary rise in V-grooved open capillaries having a small angle of aperture [14]. Additionally, the change of surface wettability will be characterized through the measurement of $\bar{h}$, the averaged front due to capillary rise, as a time function, when sandpaper of different grit is used.

Recently, the water droplets evaporation on functional surfaces has been studied, for instance, on micro-structured surfaces with hydrophilic and hydrophobic micropillars [15]. There, it has been observed that the rate of evaporation, $\dot{m}$, as a function of time $t$, of sessile droplets on both types of micropillar-structured surfaces obeys a relationship of the form $\dot{m} \propto t$, for most of the evaporation time. This can be attributed to the fact that when the droplets are sufficiently large (during the initial stages), evaporation is primarily governed by vapor diffusion at the liquid–vapor interface and heat conduction through the droplet.

The irregular texturing applied to acrylic surfaces also influences the temporal mass reduction of droplets, but the question is if evaporation or spreading dominates over one another. To quantify these processes, experiments of sessile droplets on scratched acrylic surfaces and on human skin will be also analyzed. In both of these latest cases, we will show that the droplets loss their mass at a high rate due to the capillary penetration into the V-grooves Consequently, evaporation seems to be a marginal phenomenon in the loss of mass of the droplets. The fast reduction of the mass of the water droplets on scratched surfaces, allows us to envisage that the virus SARS-CoV-2, which is carried by respiratory droplets to the functional surfaces must survive a fraction of time on scratched surfaces in comparison with its survival on smooth surfaces.

To reach our goals, in the next section we will revisit the main theoretical results of capillary action for two cases: the shape of a meniscus on a vertically standing plate and the capillary rise in open V-shaped capillaries making a small angle. Later, in Section 3, we propose a protocol for texturing acrylic plates, by scratching them up with different grit sandpaper. The surface characterization with electron microscopy of the coarse and fine scratches also is reported. Later on, we experiment how silicone oil forms menisci on smooth and textured vertical sheets and how it rises upwards on acrylic textured plates. We also compare our experimental results for the average front of rise $\bar{h}(t)$ for different types of scratches and tilt angles of the sheets with the formulas given in Section 2. In Section 4, we will perform experiments on the loss of mass of water sessile droplets on human skin and on scratched surfaces. Finally, in Section 5, we give the main conclusions of this work.

## 2. Equilibrium Profile and Dynamic Capillary Rise

### 2.1. Equilibrium Profile

When a vertical plane wall is in contact with a quiescent liquid of density $\rho$, dynamic viscosity $\mu$ and surface tension $\sigma$ and it wets the wall, a meniscus is formed under the gravity action, see Figure 1. The height of the free surface on the wall can be determined through the capillary equilibrium condition given by the balance between the Laplace and the hydrostatic pressures, it yields the equilibrium height $h$ given by [16].

$$h = l_c \sqrt{2(1 - \sin\theta)}, \tag{1}$$

where the capillary constant, or *capillary length*, is $l_c = \sqrt{\sigma/\rho g}$, $g$ is the acceleration due to gravity and the angle of contact is $\theta$, which is a property of the liquid–solid contact, meaning that a perfect wetting yields $\theta = 0°$ and then $h = \sqrt{2}l_c$.

For water at 25 °C, $\sigma = 71.97$ mN/m, $\rho = 997.04$ kg/m³, therefore $l_c = 2.71$ mm, consequently the maximum height of a water meniscus under perfect wetting is $h = 3.84$ mm.

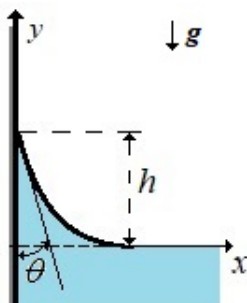

**Figure 1.** Schematic of the liquid meniscus formed, under the gravity action, on a vertical solid surface where $\theta$ is the angle of contact and $h$ is the maximum height on the wall.

### 2.2. Capillary Rise

Later, we will experimentally show that the way by which the face was scratched of the acrylic sheet with sandpaper produces a micro-textured surface with V-grooves, distributed over the entire face. As mentioned earlier, several authors [2,5,8–12] agree that the spreading of liquid on functional surfaces with V-grooves can be modeled by assuming that each groove behaves as a capillary wedge, having a small angle of aperture $\alpha$ (two plates making an small angle), which can be termed as the Taylor–Hauksbee cell [14,17,18], as the one sketched in Figure 2.

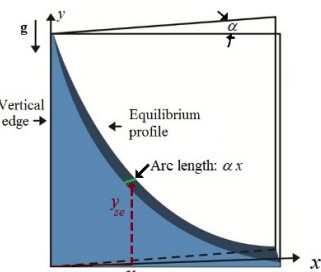

**Figure 2.** Depiction of a Taylor–Hauksbee cell, where two close together plates make a small aperture angle $\alpha$. The equilibrium profile is the last stage of the temporal evolution of the liquid free surface. Notice that at a distance $x$ from the edge, the arc length of the free surface, is approximately $\alpha x$ and there the local height of the equilibrium profile is $y_{se}(x)$.

In such a model, we assume that the lower part of a V-groove is brought into contact with a wetting liquid that will rise through the groove until it achieves a final equilibrium surface, which can be determined by knowing that the pressure jump (Laplace pressure [16]) across the surface at $x$ is approximately $\Delta p_s = 2\sigma\cos\theta/\alpha x$, where $\alpha x$ is the arc length of the free surface just at a distance $x$ from the edge. The equilibrium surface $y_{se}(x)$ of the meniscus is determined by the balance $\Delta p_s = \rho g y_{se}$ [14]. This balance yields the equilibrium surface

$$y_{se} = \frac{2\sigma\cos\theta}{\rho g \alpha x}, \tag{2}$$

which is a rectangular hyperbola whose peculiarity is that very close to the vertical edge ($x = 0$) the liquid should reach, ideally, an infinite height; however, in actual experiments, the fluid always reaches the upper limit of the plates system.

Now, we must notice that the equilibrium surface will be achieved when the temporal evolution of the free surface $y_s(x, t)$ halts, i.e., $y_s(x, t) = y_{se}$. Through the use of the Reynolds lubrication equations and the free surface evolution equation of a slow, viscous flow under the gravity action [14,19] it was found that the free surface $y_s(x, t)$ evolves in a complex manner: if one follows the elevation of the meniscus $y_s$ just at the edge, $x = 0$, for long times, the temporal evolution is as [14]:

$$y_s \approx \left( \frac{\sigma^2 \cos^2 \theta}{\mu \rho g} \right)^{1/3} t^{1/3}, \text{ at the vertical edge } (x = 0), \tag{3}$$

meanwhile, if the measurement of $y_s$ is made at $x \neq 0$, at intermediate times, the evolution of the free surface follows the power law [14]

$$y_s \approx \left( \frac{\sigma \alpha x \cos \theta}{3 \mu} \right)^{1/2} t^{1/2}, \text{ at a small distance } x \text{ from the edge.} \tag{4}$$

It is important to highlight that the power law $y_s \sim t^{1/3}$ closely fits experimental data when $\alpha$ is in the order of a few degrees and the capillary flow occurs in the edge, as schematically shown in Figure 2 [14,20]. The power law $y_s \sim t^{1/2}$, known as Lucas–Washburn law [21–23], is typical for capillary penetration in the absence of gravity into capillary tubes and porous media and in the cell it must be valid for capillary rise at small distances $x$ from the edge. All these results will be useful later to understand the way how capillary rise in functional acrylic surfaces takes place in experiments.

## 3. Experiments

### 3.1. Equilibrium Height and Characterization of the Micro V-Grooves

In our experiments, we used transparent, 3 mm thick, 80 mm height and 48 mm wide acrylic sheets. Considering that we vertically dip the sheet in silicone oil (a nonvolatile liquid at room temperature) of nominal dynamic viscosity $\mu = 100$ cP, surface tension $\sigma = 0.021$ N/m and density $\rho = 960.0$ Kg/m$^3$, formula (1) lets us find that in this case $l_c = 1.49$ mm and the height of the meniscus on the acrylic wall is $h = 1.54$ mm. In Equation (1), we used the angle of contact $\theta = 28° \pm 0.5°$, which was measured on a clean and smooth sheet, as is shown in Figure 3.

Through measurements based on numerous pictures, such as Figure 3a, we found that $h = 1.47 \pm 0.05$ mm, which is close to the theoretical value, and the difference perhaps could be caused by the meniscus on the flange of the sheet.

The other pictures of the menisci of Figure 3 correspond to (b) an acrylic surface scratched up with 150 grit sandpaper and angle of contact $\theta = 28° \pm 0.5°$ and (c) an acrylic surface scratched up with 50 grit sandpaper $\theta = 29° \pm 0.5°$. All angles of contact were measured 5400 s after each sheet was dipped into the silicone oil reservoir and, consistently, the measured angles of contact are not appreciably affected by the raw scratching.

The scratching of the acrylic sheet was carried out to produce a series of V-grooves, moreover, the use of 150 grit and 50 grit sandpapers was intended to produce fine and coarse grooves, respectively, since 150 grit corresponds to an abrasive grain size of 92 μm and 50 grit is associated to 350 μm, in agreement to ANSI (American National Standards Institute). In experiments, we carried out the scratching up with *Fandeli* [24] "wet or dry" sandpaper made of abrasive faceted silicon carbide (SiC) grains. We use this sandpaper since acrylic has a hardness of 3 on the Mohs hardness scale from 1 to 10, meanwhile silicon carbide has a hardness of 9. Acrylic surfaces are relatively soft and easily scratched by SiC grits.

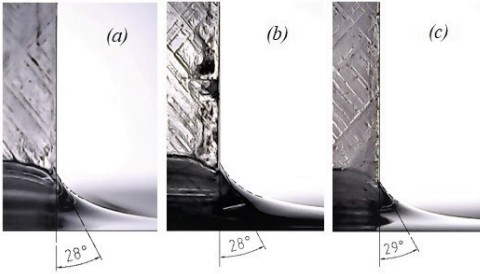

**Figure 3.** Representative pictures of menisci on vertically standing acryclic sheets, (**a**) without scratching, (**b**) scratched using 50 grit sandpaper and (**c**) scratched using 150 grit sandpaper. In all cases, the contact angles $\theta$ were measured by looking at the width of the sheet, frontally.

The protocol to produce the grooves on any acrylic face was the following: first, the face was cleaned up; then, it was uniformly and unidirectionally scratched three times (along the vertical edge); and finally, the face was cleaned up again with a fine hair brush. Following this procedure, we obtained a distribution of near parallel V-grooves. In Figure 4, we show scanning electron microscope (SEM) micrographs of specimens where the nearly parallel alignment, size and surface texture of the grooves on the acrylic face are observable. Typical width of grooves in the acrylic faces, since 150 grit sandpaper are between 10–70 μm and for 50 grit are between 100–200 μm.

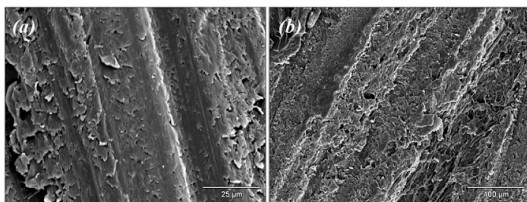

**Figure 4.** Micrographs of the near parallel grooves for (**a**) 150 grit and (**b**) 50 grit.

In Figure 5, SEM micrographs, near the edge of the sheets, where the capillary rise starts are shown for the cases where the scratching was performed with (a) 150 grit and (b) 50 grit sandpapers, respectively, notice that the V-grooves are best appreciated in the case of 150 grit, Figure 5a.

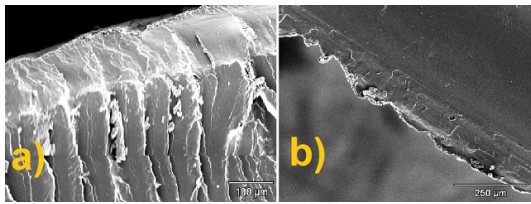

**Figure 5.** SEM micrographs near the edge of the sheets where the irregular V-grooves are observed: (**a**) 150 grit and (**b**) 50 grit.

### 3.2. Functional Surfaces: Capillary Rise

The scratching of the faces of acrylic sheets allowed us to texturize them with V-like grooves. To characterize the effect of the groove distribution on the acrylic sheets, we carried out several experiments of capillary rise. Our procedure to visualize the capillary flow along the faces consisted of dipping the sheet in a reservoir containing silicone oil 100 cP. In one case, we vertically dipped the sheet and in others tilted sheets were dipped with inclination angles of $\phi = 45°$ (counter clockwise direction) and $\phi = -45°$ (clockwise direction). Figure 6 depicts the corresponding ascending liquid films (blue thick lines) in each scratched face. In the vertical case, the flow rises opposite to gravity (Figure 6a), meanwhile in the case of $\phi = 45°$ (Figure 6b) the capillary flow surmounts the sheet, finally when $\phi = -45°$, the flow occurs at the lower face, when the liquid climbs a sloped

ceiling configuration (Figure 6c). Interestingly, in these cases the V-grooves, with different orientation, take in the liquid in very contrasting ways.

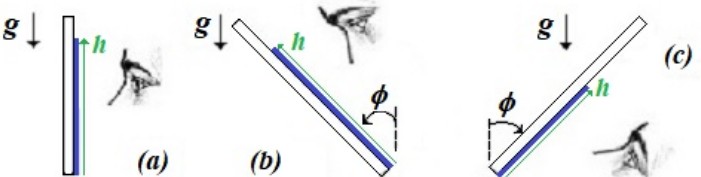

**Figure 6.** Depiction of the capillary rise of silicone oil on the functional surfaces scratched up: vertical ascent (**a**), ascent on the face tilted at $\phi = 45°$ (**b**) and ascent at $\phi = -45°$ (climbing flow at the lower face of the sheet) (**c**). The visualization of the thin films of liquid on the sheets and the measurement of respective fronts of rise is given by $h$.

In an early stage of experimentation, we dipped couples of sheets, each scratched with different grit and we video recorded, with a CCD camera, the ascent of liquid in the functional faces, as depicted in Figure 6. In Figure 7, snapshots of the capillary rise on the faces scratched with 50 grit (Figure 7a) and 150 grit (Figure 7b) are shown. On each face, the darkest lower regions correspond to well saturated zones and it is evident that the front of ascent is very irregular. To get a measure of the mean height of the front $\overline{h}$, at a given time $t$, in a second stage, we performed a simple procedure: after video recording the experiments, we obtained their single frames (digital pictures), each 1 s apart. Later, on each picture we traced ten evenly spaced parallel straight lines along the width of each face and on each line $i$ ($i = 1, \ldots, 10$) we pointed out with arrows the local position of the front which allowed us to get a measured, in mm, of the local instantaneous positions of the front $h_i(t)$; in Figure 8, we show a series of time sequential images for a sheet scratched up with grit 50 in order to observe more accurately the corresponding measures of $h_i(t)$ (in mm).

Following that, for a given time $t$, we computed the arithmetic mean of data $h_i(t)$, obtaining the instantaneous mean position of the front $\overline{h}(t)$. Such a procedure was performed for all frames of a given video recording. We carried out four different experiments for each inclination and grit, always employing a new scratched sheet in a new experiment of capillary rise. At a final stage, we again averaged the four measurements of $\overline{h}(t)$, at a given time, for each inclination of the sheet, having a given grit.

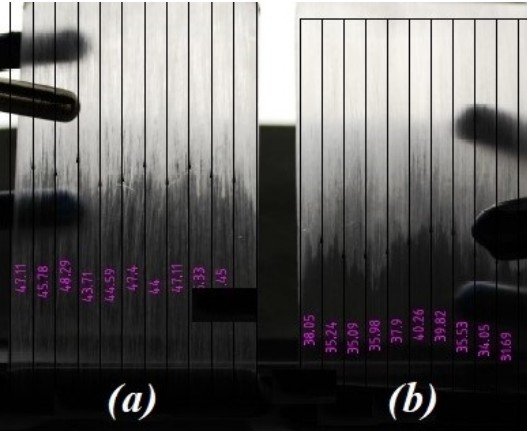

**Figure 7.** Snapshots of the capillary rise of silicone oil on a couple of vertical acrylic sheets with a face scratched up with (**a**) 50 grit and (**b**) 150 grit sandpapers. Arrowed lines, with numbers (in mm) aside, indicate the measure of the local heights, $h_i$. Notice that although the acrylic sheets were dipped at the same time, on average the height of rise is different due to the different grit on each plate.

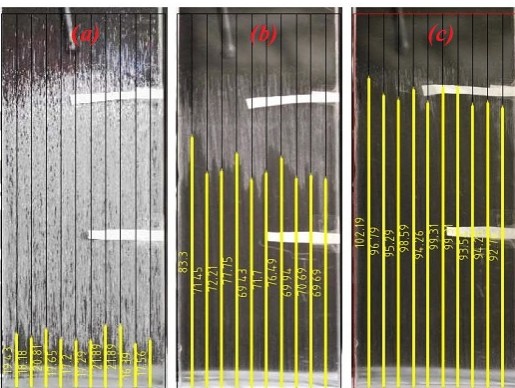

**Figure 8.** Time sequential images of the capillary rise in a vertical sheet scratched up with grit 50: (**a**) $t = 5400$ s, (**b**) $t = 170{,}220$ s and (**c**) $t = 360{,}360$ s. As in Figure 7, the numbers give the hight (in mm) of each yellow line.

In Figure 9, we show the log–log plot of the mean height $\bar{h}(t)$ on vertical sheets scratched with grits 50 and 150, respectively. There, we show that data fit the power law of the form $\bar{h} \approx at^n$, where the prefactor is $a = 0.69$ mm/s$^{0.39}$ and the exponent $n = 0.39 \pm 0.02$ for faces scratched with grit 50, and $a = 0.31$ mm/s$^{0.38}$ and $n = 0.38 \pm 0.02$ for faces with grit 150. In the inset of this figure, we show the plot of the mean speed of the fronts $d\bar{h}/dt$ (obtained from the power law fits) and this confirms that the liquid front is slightly faster in the face scratched up with 50 grit sandpaper. Is important to notice that in both cases, in the last few moments, the speed of the fronts remains nearly constant, which could be mainly attributed to flow occurring in the close vicinity of corners, since in this region the capillary pressure that drives the flow is large [14].

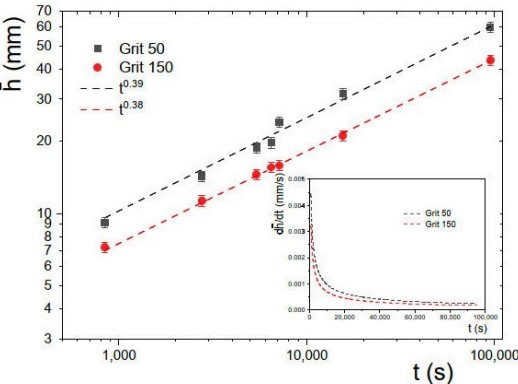

**Figure 9.** Log–log plot of the averaged height of capillary rise, $\bar{h}$, as a time function, on faces of acrylic with grit 50 and grit 150, respectively. In both cases, data fit power laws of the form $\bar{h} \sim t^n$, where $n = 0.38$ for grit 50 and $n = 0.39$ for grit 150.

With regard to the theoretical results of the capillary rise in very tight wedges discussed in Section 2, we highlight that for vertical sheets and tilted sheets with $\phi = 45°$, the average front obeys approximately that $\bar{h} \sim t^{0.40}$, which could be interpreted as an intermediate power law among that corresponding to the flow just at the edge $\bar{h} \sim t^{0.33}$ (Equation (2)) and the other corresponding to capillary rise at a small distance from the edge where $\bar{h} \sim t^{0.50}$ (Equation (3)). The same idea can be assimilated reasoning that due to our method of visualization of the fronts on the vertical sheet and on the 45° tilted sheet, we actually look at an depth-averaged flow composed of the edge flow and flow at a small distance from the edge. On the contrary, when we visualize the climbing flow on the sheet tilted at the angle $\phi = -45°$, the main flow occurs at a small distance from the edge, since gravity always tries to pull the liquid outside the V-grooves, thus in agreement with Equation (3) the exponent will have a value close to $n = 0.50$.

It can be seen that, in plots of Figures 9 and 10, there is a consistent difference between data for capillary rise in faces scratched with grit 50 and grit 150; however, they have nearly the same trend, given by the corresponding exponent $n$, consequently, the prefactor $a$ must determine such a difference. Typically, the values of $a$ for grit 50 are larger than those for grit 150. In order to explain this, we observe that the prefactor in Equation (4) contains the term $\alpha x$ (the arc length of the free surface of the liquid, located at distance $x$ from the inner edge) which in general, for grit 50 is larger than for grit 150, since the V-grooves for grit 50 have a larger depth than for grit 150; additionally, we must remember from Figure 3 that no appreciable changes of the angles of contact $\theta$ were measured (by using the method of the meniscus under gravity) for smooth or scratched surfaces. However, we must notice that this latest condition will not be maintained for the sessile droplets on functional surfaces, as will be discussed in the next Section.

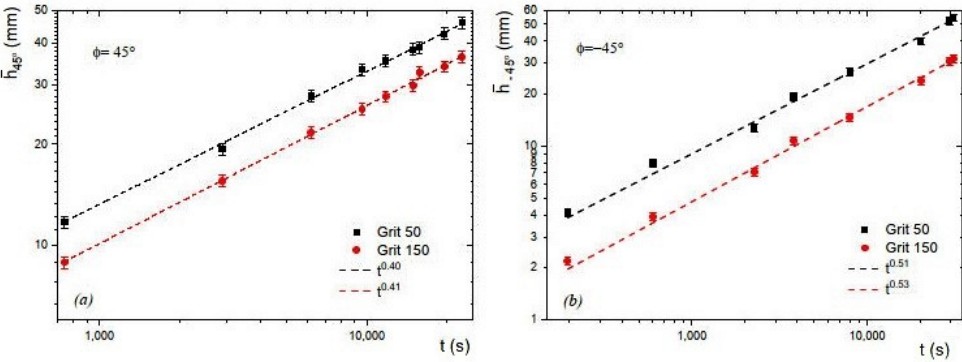

**Figure 10.** Log–log plot of the averaged front position, $\overline{h}$, as a time function, on faces tilted at (**a**) $\phi = 45°$ and (**b**) $\phi = -45°$, respectively. In both cases, we show data for 50 and 150 grit scratching. When $\phi = 45°$ data approximately fitted power laws of the form $\overline{h} \sim t^n$, where $n = 0.40$ for grit 50 and $n = 0.41$ for grit 150. Similarly for case $\phi = -45°$ data fitted power laws with $n = 0.51$ for grit 50 and $n = 0.53$ for grit 150.

All this quantitative characterization of the functional surfaces, specifically through the statics and dynamics effects of the capillary action, shows that our approximate models of fluid mechanics are very sensitive tools for these types of disordered or complex patterns. Now, the question is if a similar approach can be useful for the characterization of another phenomena on functional surfaces with irregular V-grooves, it will be studied experimentally later on.

## 4. Functionality of Scratched Surfaces in Other Cases

### 4.1. Droplets on Human Skin

We must notice that in nature and technology there are a myriad of functional surfaces following countless patterns. For instance, in Figure 11 we can observe that human skin has different microtextures on its different parts: in Figure 11a, we show the skin groove network on the forearm, in Figure 11b, the grooves on a fingertip are shown, whereas in Figure 11c, a water drop hanging from the forefinger can be visualized. In the case of spreading of liquid on a human forearm, the rate of spreading of a drop of a moisturizing oil was visualized through the fine surface V-grooves (characterized through replicas), obtaining that its advance front, $L$, follows a power law of the form $L \sim t^{0.5}$ [12]; to our knowledge the spreading of liquids on human forefingers is still an open problem, despite the well-known record of human fingerprints.

Similarly, the problem of the shape of a drop hanging from the fingertip depends on the random distribution of grooves and is awaiting for an accurate treatment. The skin is a live organ and is a tough outer protective layer that keeps water repellency, our proposal is that novel materials, using replicas, micromachining or directed sanding, may be useful to replicate geometrical configurations that nature has printed for eons to understand functionalities of our human integument.

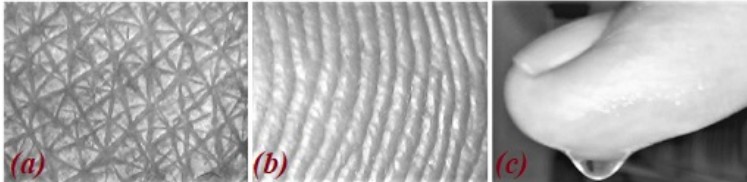

**Figure 11.** Pictures of human skin groove networks of (**a**) forearm skin, (**b**) a fingertip and (**c**) a water drop hanging from the forefinger. Sample sizes in (**a**,**b**) are of a few millimeters.

Now, results interesting to follow the loss of mass of a water sessile droplet on skin; for instance, in Figure 12 we show three snapshots of a water droplet on the dorsal wrist skin, we observe that the droplet loss mass and we characterized this phenomenon by measuring the angle of contact $\theta$ as a function of the elapsed time, $t$. The plot of $\theta(t)$ in Figure 12d indicates that at short times the change of $\theta$ is very fast and at the last stage, the rate of change diminish. We will show later that such behavior is characteristic of sessile droplets on V-grooves.

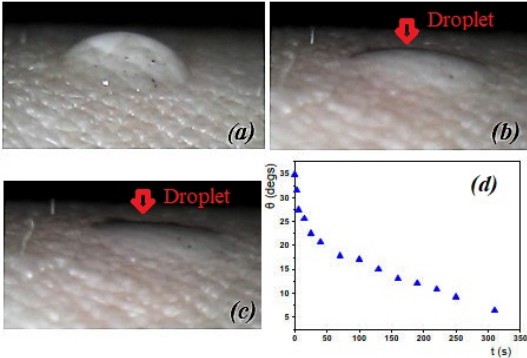

**Figure 12.** Time sequential images of the spreading of a water droplet on dorsal wrist skin: (**a**) $t = 0$ s, (**b**) $t = 163$ s, (**c**) $t = 198$ s and (**d**) plot of the change of the contact angle $\theta$ as a function of time, $t$.

### 4.2. Droplets on Scratched Acrylic Sheets

Finally, another functionality of acrylic was experimentally tested here: the loss of mass of water sessile droplets on scratched surfaces. This case is important due to its possible relevance on the surface transmission of SARS-CoV-2, the virus that causes COVID-19. The principal mode by which people are infected is through exposure to respiratory droplets carrying the infectious virus. SARS-CoV-2 is an enveloped virus, meaning that its genetic material is packed inside an outer layer (envelope) of proteins and lipids. The envelope contains structures (spike proteins) for attaching to human cells during infection. Since the droplet serves as a medium for virus survival, the infectivity of the virus is connected to an extent to the droplet lifetime.

Researchers have studied how long SARS-CoV-2 can survive on a variety of porous [25,26] and smooth surfaces [26,27]. On porous surfaces, studies report an inability to detect a viable virus within minutes to hours; on smooth surfaces, viable virus can be detected for days to weeks. The apparent, relatively faster inactivation of SARS-CoV-2 on porous compared with smooth surfaces might be attributable to capillary action within pores and faster aerosol droplet evaporation.

In Figure 13, we show the temporal evolution of a water droplet of approximately 1 mm diameter, whereas it evaporates on a smooth acrylic surface, we found that the time of evaporation under these conditions is of around 12 min. Similarly, we placed a water droplet of 1 mm on an acrylic surface with cross scratching (grit 100 sandpaper), see Figure 14. Now, the total loss of mass, on such a surface took place in about 3 min, in this case we observed that the main reason for it is the capillary penetration of liquid, into the horizontal micro V-grooves.

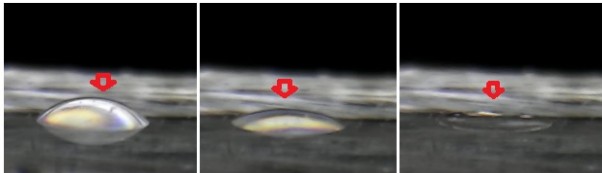

**Figure 13.** Snapshots of evaporation of a water drop (pointed out with arrows) on a smooth acrylic sheet. In this case evaporation took around 12 min.

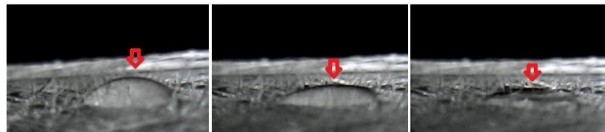

**Figure 14.** Snapshots of spreading of a water droplet (pointed out with arrows) on an acrylic sheet, which was cross scratched with a 100 grit sandpaper. In this case, the total loss of mass took around 3 min.

To highlight the importance of the type of scratching of the acrylic sheets, and its effect on the loss of mass of the droplets, in Figure 15 we show snapshots of the spreading of a water sessile droplet on a longitudinal scratching (grit 150), obtained by using the same protocol of Section 3. At time $t = 0$ the droplet is near circular, from the top view, after, at $t = 2$ s the droplet increases its diameter due to the high wettability between acrylic and water (the contact angle is $\theta_w = 36.5° \pm 1°$). A few seconds later, the mass of water flows spontaneously, due to the capillary action, along the V-grooves. Finally, there occurs the formation of very small drops, perhaps due to the instabilities in the fast flow, typical of the open V-shaped microchannels [28].

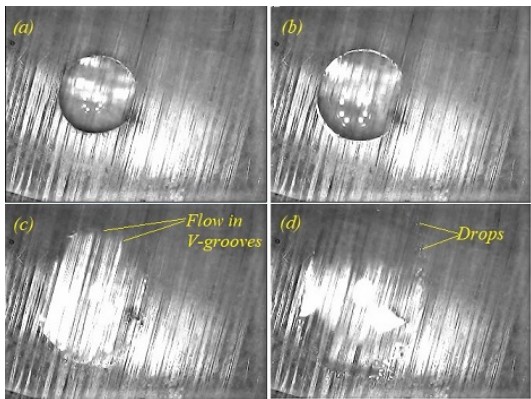

**Figure 15.** Snapshots of the top view of a water sessile water droplet of initial diameter $D \approx 1.5$ mm on an acrylic sheet with longitudinal scratching (grit 150): (**a**) $t = 0$ s, (**b**) $t = 2$ s, (**c**) $t = 34$ s and (**d**) $t = 87$ s. In (**c**), the capillary flow along the open V-grooves is appreciated, and in (**d**), the formation of very small drops there occurs.

The previous description details how a water droplet, on scratched surfaces, lose their mass, mainly due to the spontaneous capillary flow along the open microchannels. To reinforce these ideas, we also measured the change of the apparent contact angle, $\theta'$, of sessile droplets, as a function of time.

Studies of the droplet evaporation dynamics on hydrophobic or hydrophilic micro-pillared surfaces give plots of the contact angle of sessile water droplets [15]. In both cases, droplets evaporating on a structured surface predominantly exhibit the droplet contact angle decreasing slowly as the droplet evaporates and, after, the decreasing is faster. The first is attributed to the fact that when the droplets are sufficiently large (during the initial stages), evaporation is primarily governed by vapor diffusion at the liquid–vapor interface and heat conduction through the droplet.

To contrast with the previous case, in Figure 16, we show the plots of the apparent contact angles $\theta'(t)$, for water and silicone oil sessile droplets on longitudinal scratching acrylic sheets. It is appreciated that the change of these angles is very similar among them, despite silicone oil is a nonvolatile liquid at room temperature. Moreover, the contact angle of a water droplet on skin, given in Figure 12, changes in a similar manner. Contrary to cases of droplets on micro-pillared surfaces, all cases of loss of mass of droplets on V-grooved surfaces presents, during the initial stages, a strong loss of mass, quantified through the angles $\theta'(t)$ or $\theta(t)$, respectively. At the final stages, the slow loss of mass also occurs for all cases.

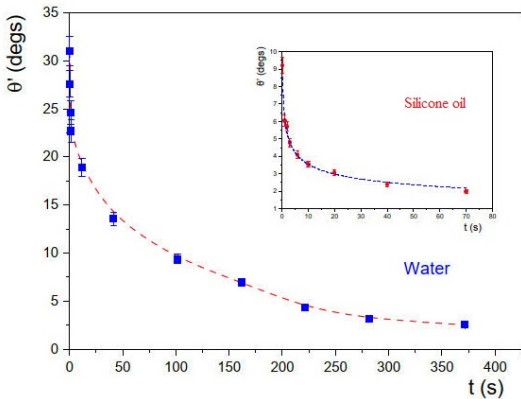

**Figure 16.** Plot of the temporal evolution of the apparent contact angle $\theta'(t)$, for a water sessile droplet and for a silicone oil sessile droplet (inset). The diameter of both droplets were $D \approx 1.2$ mm and it is below of the respective capillary lengths of water and silicone oil, see text.

We advice that measurements of $\theta'$, for times lower than 0.5 s, were not possible due to limitations of our microscope, however the static measurement of the contact angle for a sessile drop of silicone gave the value $\theta = 26° \pm 1°$, which is close to the contact angle measured through the method of the meniscus under gravity given in Section 2.

Our experimental observations on the spread of droplets on longitudinal scratching surfaces allows us to conclude that the fast loss of mass is due mainly to the capillary flow into the series of parallel open V-grooves which sustain the droplet, which also could be interconnected, meanwhile the slow loss of mass, at the later stages, it is imposed by the strong influence of the viscous stresses along the same series of V-grooves. The evaporation of the liquids in the horizontal V-grooves has also been reported [28,29].

In summary, in a general context, many solid surfaces suffer a profuse quantity of scratches, but in the context of COVID-19, when a droplet carrying the infectious virus is deposited on a scratched surface, it rapidly loses its envelope aqueous layer, and thus its survival time is very short. So, the scratching of surfaces brings up a beneficial functionality.

## 5. Conclusions

In this work, we experimentally studied the functionality of acrylic sheets when they were subjected to irregular scratching, nearly along a single direction. In a first stage, we revisited the classical theory of the capillary action by showing the formation of a liquid meniscus on vertically standing acrylic sheets. Afterwards, in the same context of the capillary action, we highlighted the various power laws, depending on the place of measurement, of capillary rise when a viscous and wetting liquid spontaneously penetrates into a Taylor–Hauksbee cell. Taking into account all these fundamental features of capillary action, we used 50 and 150 grit sandpapers in order to carve V-grooves on the face of the acrylic sheet and several of their geometrical characteristics were explored with the use of electronic microscopy.

An initial analysis of the functionality of the scratched surfaces was performed through the study of the capillary rise in vertical and tilted sheets, where measurements of the non-uniform fronts of silicone oil, for each case, were obtained. It is apparent that the scratching

of surfaces with grit 50 produces larger grooves in comparison with those produced with grit 150. It was found that the averaged fronts of rise, $\bar{h}(t)$, obey power laws of the form $\bar{h}(t) \approx at^n$, with approximate values $n = 0.40$ for vertical and tilted sheets when $\phi = 45°$, meanwhile for the case when $\phi = -45°$ we found that approximately $n = 0.50$. Thus, the use of power laws given by Equation (2) or (3) is not direct since the size and orientations of the grooves affects the value of the exponent $n$, meanwhile the prefactor $a$ depends more on the properties of the involved liquids and the grit size.

We also have argued about the possible utility of our procedure to study, as a geometric analog, the functionality of the human skin since our skin has different types of grooves at different parts of our body. It means that there are an infinity of functional characteristics in the human skin and it is possible that different types of scratches on acrylic (linear, circular, crossed, etc.), or types of micro patterning, can emulate simple micro flows on skin. Moreover, experiments of the loss of mass of sessile water droplets on skin allowed us to understand the importance of the V-grooves because they suck out the water from the droplets themselves. At the end of the current paper, we also gave evidence of the functionality of scratched surfaces for sessile water droplets because they loss mass very rapidly. We proved, for several fluids, that the loss of mass of sessile droplets is mainly due to the strong capillary flow in the open V-grooves. It allows us to conclude that in case that aqueous droplets containing SARS-CoV-2, fall on dry scratched surfaces, the time of loss of their aqueous layer is shorter than that of a smooth surface. As a consequence, the coronavirus remains viable a shorter time on scratched surfaces compared to smooth surfaces, a similar conclusion for many porous surfaces was found by other authors.

In broad context, wear of surfaces due to continuous use and cleaning, and specially when abrasives substances are used, provokes involuntary scratching that finally has a functionality as the one found in our controlled study, moreover, we hope that further studies will improve our comprehension of scratched surfaces.

**Author Contributions:** Experiments A.M., A.L-V., C.A.V.; modeling: A.M., A.L-V., C.A.V.; writing and revision: A.M., A.L-V., C.A.V. All authors have read and agreed to the published version of the manuscript.

**Funding:** This research received no external funding.

**Data Availability Statement:** The data presented in this study are available on request from the corresponding author. The data are not publicly available due to we still do not have a publicly accessible repository.

**Acknowledgments:** A.M. acknowledges F.J. Higuera, S. de Santiago, A. Jara, R. Diéz-Barroso, J. Casillas, D.A. Serrano and F. Hernández-Santiago their help in the different phases of this work.

**Conflicts of Interest:** The authors declare no conflict of interest.

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
