# Peer review of "Functional Acrylic Surfaces Obtained by Scratching"

_fluids, doi:10.3390/fluids6120463_

Round 1

Reviewer 1 Report

In this article, the authors experimentally studied the functionality of acrylic sheets with irregular but single-direction scratching. The structure and logic of the article are clear and sound. The major issue for me is that some of the long sentences are hard for me to interpret quickly. Please see my review comments below. 

Major

  1. Since the infectivity of the virus can be related to the droplet evaporation on the surface, a brief literature review can be added to the Introduction in terms of the droplet evaporation on the functional surfaces.
  2. Figure 2 (a) was very confusing to me at first. I had to check reference 13 and then realized that Figure 2 (a) is actually depicted in 3D coordinates. Improvements can be made to Figure 2 (a) so that readers can easily interpret it.
  3. Line 92-93, are there any references for Reynolds lubrication equations and the free surface evolution equation of a slow and viscous flow?
  4. The authors mentioned that "the size and orientations of the grooves affects the value of the exponent n" in line 283, so Eqs. (2) and (3) cannot be used directly. Is gravity considered in the derivation of Eqs. (2) and (3)?

Minor

  1. Line 9, typo "rol of weared surface".
  2. Line 13, please check the grammar.
  3. Line 17, missing "are" after "essential".
  4. Line 55, missing a conjunction word.
  5. Line 62, should "it" be "its"?
  6. Line 103-104, please check the grammar.
  7. Line 207, what does "said" mean here?

Reviewer 2 Report

The reviewer finds that the paper presents some originality in th approach. The background is helpful to better understand the results obtained experimentally on micro-grooves.

The results on the statics and dynamics of capillary action in disordered patterns is interestingly related to nature made functionalities.

Reviewer 3 Report

Review Comments for fluids-1459897

Functional acrylic surfaces obtained by scratching

by Abraham Medina, Abel López-Villa, Carlos A Vargas

This manuscript experimentally investigated the capillary phenomenon on functional acrylic surfaces obtained by scratching. A power law relating the height and time was established. Although some results were reported, plenty of issues need to be addressed before considering for publication. The following are the detailed comments.

  1. Line 50-53, did this paragraph also introduce the background? So, it should be moved before Line 36.
  2. Figure 3, How did the contact angles measured by capillary rise differ from those by sessile droplets? Generally, a sessile droplet is used to measure the contact angle of a surface and a rougher surface is more hydrophilic or hydrophobic according to the Wenzel equation (Murakami, et al. Langmuir 2014, 30 (8), 2061-2067.).
  3. Line 121-141, the preparation for experimental surfaces should be moved before Figure 3.
  4. Line 136-137, could the authors give the surface profiles to quantitatively compare the different surface characteristics and ensure the repeatability of the prepared surfaces?
  5. What is the meaning of the number in Figure 7? Could the authors supplement the time-sequential images?
  6. Line 209 – 212, what is the physical meaning of αx? “Additionally, we must remember from Figure 3 that no appreciable changes of the angles of contact θ were measured for”, pls reconsider Comment 2.
  7. Could the authors conduct a quantitative comparison of the droplets on human skin and functional surfaces?
  8. Figure 12, pls indicate the time scale and present the temporal evolution of droplet radius or height during evaporation.

Round 2

Reviewer 3 Report

The revised manuscript could be accepted for publication.